# Amelioration of Brain Damage after Treatment with the Methanolic Extract of *Glycyrrhizae Radix et Rhizoma* in Mice

**DOI:** 10.3390/pharmaceutics14122776

**Published:** 2022-12-12

**Authors:** Myeongjin Choi, Chiyeon Lim, Boo-Kyun Lee, Suin Cho

**Affiliations:** 1Department of Korean Medicine, School of Korean Medicine, Pusan National University, Yangsan 50612, Republic of Korea; 2Department of Medicine, College of Medicine, Dongguk University, Goyang 10326, Republic of Korea; 3Department of Radiology, Massachusetts General Hospital and Harvard Medical School, Charlestown, MA 02129, USA

**Keywords:** licorice, ischemic stroke, glycyrrhizic acid, cerebral infarct

## Abstract

*Glycyrrhizae Radix et Rhizoma* (GR) is a traditional herbal medicine widely used in Asian countries. GR was the most frequently used medicine among stroke patients in Donguibogam, the most representative book in Korean medicine. In the present study, we investigated the neuroprotective effects of the GR methanolic extract (GRex) on an ischemic stroke mice model. Ischemic stroke was induced by a 90 min transient middle cerebral artery occlusion (MCAO), and GRex was administered to mice with oral gavage after reperfusion of MCA blood flow. The MCAO-induced edema and infarction volume was measured, and behavioral changes were evaluated by a novel object recognition test (NORT). Immunofluorescence stains and Western blotting identified underlying mechanisms of the protective effects of GRex. GRex post-treatment in mice with MCAO showed potent effects in reducing cerebral edema and infarction at 125 mg/kg but no effects when the dosage was much lower or higher than 125 mg/kg. GRex inhibited the decrease of spontaneous motor activity and novel object recognition functions. The neuroprotective effects of GRex on ischemic stroke were due to its regulation of inflammation-related neuronal cells, such as microglia and astrocytes.

## 1. Introduction

Stroke is a general term for the symptoms of localized neurological deficits caused by abnormalities in cerebral blood flow. Managing patients suffering from stroke disability is of great medical interest because stroke causes permanent disability, and the medical costs associated with patient management are high [1,2]. Stroke is a term for symptoms, and when it is called a medical disease, it is called a cerebrovascular accident (CVA). The incidence of CVA is increasing due to the spread of diseases, such as obesity, diabetes, and the elderly population. The age of onset tends to decrease due to changes in diet and lifestyle [1,2,3,4].

In the Asian traditional medical community, especially in Chinese and Korean medicine, stroke is referred to as a ‘wind strike’. However, the scope of traditional medicine in treating stroke symptoms is broader than in Western medicine; for example, the ‘wind strike’ includes diseases such as facial nerve palsy [5,6]. Therefore, it can be seen that the method of classifying and treating stroke in traditional medicine is different from that of Western medicine.

Asian medicine originated in China and had a history of over 2000 years. Today, it consists of several medicines mainly practiced in East Asia, such as traditional Chinese and Korean medicine. A unique medical system of traditional Korean medicine was established after Donguibogam was published in 1613 [7]. In 2009, the United Nations Educational, Scientific, and Cultural Organization inscribed the book on its Cultural Heritage List in light of its historical medical value.

Donguibogam occupies an essential position in Korean medicine and contains various clinical experiences, and many prescriptions in this book are frequently used by doctors of Korean medicine. However, objective evidence to support this use is still lacking. Recently, we reported that *Glycyrrhizae Radix et Rhizoma* (GR) was most frequently used in the prescriptions for ‘wind’-related symptoms in the contents of Donguibogam, and it was confirmed that GR was blended in 55 types of prescriptions out of 89 types of complex prescriptions used for ‘wind’ [8]. ‘Wind’-related diseases in Korean medicine are partially classified as diseases related to stroke in Western medicine [5,6]; therefore, in this study, GR, the most frequently used medicine for treating stroke-related symptoms in Donguibogam, was selected as the research material.

GR, also known as licorice, has been used to regulate the action of other medicines. Over 20 species from the *Glycyrrhiza* genus, including *G. uralensis*, are commonly used in Asian traditional medicine [9,10,11,12,13]. In Korean medicine, GR is commonly used for helping the pharmaceutical action of other medicinal substances, so the various effects of GR have been underestimated; thus, research on the single compounds isolated from GR is more active than the form of GR crude extract. Several phytochemicals isolated from GR have neuroprotective effects, including anti-inflammatory, anti-excitotoxic, and anti-apoptotic pathways [14,15,16]. A study has found that glycyrrhizic acid is effective in cerebral infarction in mice [17]. We have reported that GR pretreatment showed neuroprotective effects via an anti-apoptotic pathway in a mouse model [18]. However, a study on the post-treatment of GR in a rodent model of ischemic stroke has not yet been published.

Therefore, in this study, we evaluated that GR could be effectively used for cerebrovascular disease through network pharmacological analysis, and it was confirmed that the methanolic extract of GR (GRex) was effective in middle cerebral artery occlusion (MCAO)-induced brain injury in mice.

## 2. Materials and Methods

### 2.1. Screening of Active Substances from GR and Target Diseases through Network Pharmacology Analysis

Using Traditional Chinese Medicine Systems Pharmacology (TCMSP), a database for systems biology research, the bioactive substances of GR were selected in consideration of absorption, distribution, metabolism, and excretion, provided by the TCMSP [19]. Potential active substances were selected using screening conditions, including oral bioavailability (OB), intestinal epithelial permeability (Caco-2), blood–brain barrier (BBB), and drug-likeness (DL) among variables that could be considered as the characteristics of this study. The variables used in this study were set as threshold values of OB ≥ 20%, Caco-2 ≥ 0, BBB ≥ −0.30, and DL ≥ 0.18. Diseases predicted to act on each component selected above were also selected using TCMSP. The screened compounds and the network with diseases classified as these compounds were visualized using Cytoscape v3.9.1 (Cytoscape Team).

### 2.2. Preparation of GRex

The GR used in this study was obtained from a professional pharmaceutical company (Gwangmyoung Pharmaceuticals Co., Gwangmyeong, Korea). The above company verified that this GR sample met the Korea Ministry of Food and Drug Safety criteria to be used as herbal medicine and was processed from plants originating from G. uralensis. In order for GR to be used as a medicine and according to the Korean Pharmacopoeia, GR should contain 2.5% or more of glycyrrhizic acid (C_42_H_62_O_16_: 822.93) and 0.7% or more of liquiritigenin (C_15_H_12_O_4_: 256.27) for the converted dry matter when quantified [20]. The voucher specimens (No. 19GR-2483) were deposited at the Herbarium of School of Korean Medicine, Pusan National University.

To prepare GRex, 200 g of GR was immersed in 2000 mL of methanol at 25 °C for three days, then filtered using filter paper (Whatman No. 1, WhatmanTM, Maidstone, UK), and the supernatant was separated. Then, 1000 mL of methanol was added to the filtered GR residue for two days, and the mixture was filtered again. The supernatant was collected twice and concentrated under reduced pressure (SB-1000, Sunil-Eyela, Seongnam, Korea) and freeze-dried (FD, Ilshin BioBase, Dongducheon, Korea). The weight of the GRex obtained was 15.4 g (yield: 7.7%). GRex was stored in 50 mL conical tubes at −20 °C until use.

To validate the quality of GRex, we determined the presence of glycyrrhizic acid, a major component of GR, using an HPLC system (YL-9100, Young In Chromass, Anyang, Korea). The samples were separated using a ZORBAX Eclipse XBD-C18 column, and the data are shown in Figure 1A. The amount of glycyrrhizic acid contained in the GRex was found to be 3.1%, and it satisfied the criteria for GR in the Korean Pharmacopoeia [20].

### 2.3. Animals

Male C57BL/6 mice aged 8 weeks and weighing 23–25 g were obtained from a specialized company in the production of laboratory animals (Samtako Bio, Osan, Korea) and adapted to the laboratory environment at least 1 week before use. All animals were housed in an environment at 22 ± 1 °C temperature and 12 h light/dark cycle and were fed standard rodent chow and provided water ad libitum. The ethics committee approved all experimental protocols of the Pusan National University (Approval No. PNU 2019-2483).

The experimental groups included the sham-operated (sham) group, MCAO-operated model (model) group, and GRex treatment groups with several concentrations, each containing at least five animals per experimental group. All procedures, including surgical operations, were approved by the Pusan University Animal Experimental Ethics Committee, and each experimental procedure was conducted in compliance with related provisions.

### 2.4. Administration of GRex

On the day of administering GRex to animals, GRex was dissolved entirely in dimethyl sulfoxide (DMSO), diluted in physiological saline, and orally administered. The final administration volume was adjusted to 5 mL/kg body weight. GRex was administered after the surgical procedure was completed. The normal and model groups were orally administered the equivalent amounts of DMSO and physiological saline.

### 2.5. Mice MCAO Modeling

In the model and GRex-treated groups, surgery for MCAO modeling was performed as previously described [21] with some modifications. Anesthesia was induced by inhalation of 2% isoflurane in 70% N_2_O and 30% O_2_, and body temperature was maintained at 36.5 ± 0.5 °C using a thermostatic blanket (55-7020, Harvard Apparatus, Holliston, MA, USA) with an attached thermometer. The chest and neck hair was removed with clippers, and an incision was made in the middle of the neck. The branches of the left common carotid artery (LCCA), external carotid artery, and internal carotid artery were carefully separated from the surrounding connective tissue. During surgery, the external and common carotid artery (CCA) was ligated with sutures to block blood flow. The blood flow in the internal carotid artery was temporarily blocked. In this state, a silicon-coated nylon suture (8-0 monofilament, Ethicon, Edinburgh, UK) 11 mm in length was gently inserted from the internal carotid artery to the left MCA origin. MCAO was confirmed by a decrease in relative cerebral blood flow (rCBF) in MCA to 20% of normal; rCBF was measured with a laser Doppler flowmeter (moorVMS-LDF, Moor Instrument, Axminster, UK). Briefly, the optical fiber was firmly fixed with surgical glue to the skull where the left MCA was located, and the rCBF before LCCA ligation was considered 100%.

The inserted filament was fixed to the proximal area of MCA for 90 min to maintain MCAO, the filament was carefully pulled out to restore blood flow, and reperfusion was performed for 22.5 h. The skin was sutured, and each mouse was awakened from anesthesia. The sham group performed a sham operation by ligating the CCA and suturing the dissected muscle and skin. A schematic view of GRex administration, induction of ischemia and perfusion, and sacrifice of experimental animals are shown in Figure 1B,C.

### 2.6. Analysis of Damaged Area of Brain Tissue

Twenty-four hours after starting MCAO, the experimental animals were euthanized with CO_2_. Brains were excised, and to keep the brain tissue fresh, the excision was performed on ice, and the excised brains were sectioned at 1 mm thickness using a mouse brain matrix (Kent Scientific, Torrington, CT, USA) and stained with a solution of 2% 2,3,5-triphenyltetrazolium chloride (TTC). The sections were fixed in 10% formalin for at least 2 h and photographed with a digital camera. TTC stains viable tissue red, while necrotic areas remain white. Each section’s cerebral edema and infarct area were quantified and analyzed using image analysis software (Digimizer, Ostend, Belgium). The area increased due to cerebral injury was measured for cerebral edema, and cerebral infarction was expressed as the rate of infarct area occurring in the ipsilateral region [22,23].

### 2.7. Neurobehavioral Evaluation

To determine the extent of motor performance impairment in rodent models of stroke, neurological deficit scores (NDS) were assessed 24 h after MCAO application, and the extent of impairment was quantified using a 5-point scale as follows:

Grade 0: No neurological abnormalities.

Grade 1: Spontaneous movement can be performed smoothly, but the surroundings cannot be grasped smoothly with the front legs.

Grade 2: Locomotor activity is relatively easy, but when the mouse is pulled by its tail and placed on a flat bottom, the body shifts in one direction (right or left).

Grade 3: Walking or turning in one direction when stimulated, animals are nociceptive to stimulation of the tail.

Grade 4: No locomotor activity, susceptible to pain when the tail is stimulated.

### 2.8. Forepaw Grip Strength Test

After completing the NDS test, mice were placed on a wire grid attached to a custom-made grip test device, grabbed the grid with both front paws, and then slowly pulled until the grip was released. The maximum force generated was recorded in grams. All tests on each mouse were performed in triplicate, and the average value was regarded as the measured value.

### 2.9. Novel Object Recognition Test (NORT) and Spontaneous Motor Activity Measurement

NORT is used to assess the neophilic propensity of rodents to new objects compared to familiar objects. During the adaptation phase, each mouse was allowed to explore the open-field arena (40 × 40 × 40 cm gray box) for 5 min without objects. In the first trial, two identical objects were placed in the two opposite corners of the test arena, and the animals were allowed to explore the two objects for 10 min. After 20 min, mice were placed in the arena again in the second trial, and one of the identical objects provided in the first trial was replaced with a novel object. The time spent exploring each object was recorded for 10 min. Spontaneous motor activity was evaluated by measuring the total distance in the arena for 10 min. Analysis of objects’ search time and discrimination ratio (DR) was calculated as the ratio of the time spent in the zone with the novel object to the time spent in the zones with the two objects. The movements of the experimental animals in the arena were recorded with a digital camera and then analyzed using video tracking software (SMART, Panlab Harvard Apparatus, Holliston, MA, USA). Objects and the arena were cleaned with 70% ethanol between trials.

### 2.10. Cardiac Perfusion and Brain Cryosectioning

Mice were sacrificed by CO_2_ inhalation and subjected to cardiac perfusion/fixation using PBS containing 4% paraformaldehyde (PFA). The fixed brains were immersed in 4 °C PBS containing 10%, 20%, and 30% sucrose sequentially for cryoprotection and cryosectioned at 25 µm using a cryostat (CM3050 S, Leica, Wetzlar, Germany).

### 2.11. Immunofluorescence (IF) Stain

The sections were dried on a slide warmer and incubated with blocking buffer (5% BSA) for 1 h at 25 °C, washed with blocking buffer. Diluted primary antibodies against neuronal nuclear (NeuN), glial fibrillary acidic protein (GFAP), the cluster of differentiation 68 positive (CD68+) (Cat. no. 94403, 12389, and 97778, respectively (Cell Signaling Technology, Danvers, MA, USA), and tumor necrosis factor-alpha (TNF-α) (ab1793, Abcam, Cambridge, UK) were incubated with the sections overnight at 4 °C. The primary antibodies were washed three times with PBS for 5 min, and the diluted secondary antibodies (goat anti-mouse or goat anti-rabbit IgG H&L, Abcam, Cambridge, UK) were dropped on the sections at 25 °C for 2 h. The secondary antibodies were washed three times with PBS for 5 min each. After dropping the mounting medium with DAPI (ab104139, Abcam, Cambridge, UK), the sections were covered with cover slides, and the edges were sealed with nail polish. After observation under a fluorescence microscope (Ni-U, Nikon, Tokyo, Japan), the samples were stored in a refrigerator at 4 °C for long-term storage. The degree of color intensity in response to fluorescence after staining with an antibody was regarded as the expression level of the protein, and the fluorescent intensity was analyzed using ImageJ (NIH, Bethesda, MD, USA).

### 2.12. Western Blot Analysis

Experimental animals were euthanized with CO_2_ 24 h after MCAO was initiated, and the brains were excised from the skull and placed on ice. The left hemisphere of the excised brain, including the putative penumbra, was isolated and incubated in a lysis buffer containing 1% Triton X-100, 0.32 M sucrose, 5 mM EDTA, 1 mM DTT, and 10 mM Tris (pH 7.4). After removing the residue with a centrifugal separator, the protein concentration was quantified.

Equal amounts of protein were dissolved in the sample buffer, and proteins in the lysis buffer were denatured by heating in a heating block for 5 min. Proteins were separated by 10% SDS-polyacrylamide gel electrophoresis and transferred to nitrocellulose membranes (WhatmanTM, Maidstone, UK). Membranes were blocked with 5% skimmed milk in tris-buffered saline containing Tween (TBST) buffer for 1 h at 25 °C. Membranes were then incubated with primary antibodies against nuclear factor kappa B (NF-κB), phospho (p)-NF-κB, and toll-like receptor 4 (TLR4) (1:1000 dilution) or β-actin (1:2000 dilution) for 12 h at 4 °C. Membranes were then incubated with horseradish peroxidase (HRP)-conjugated goat anti-rabbit IgG, pAb (1:5000) or HRP-conjugated goat anti-mouse IgG pAb (1:3000) for 2 h at 25 °C.

Immunodetection was performed and quantified using a photoluminescence spectrometry system (Amersham™ Imager 600, Piscataway, NJ, USA) according to the protocol provided by the manufacturer. All bands were analyzed using ImageJ (NIH, Bethesda, MD, USA).

### 2.13. Statistical Analysis

All analyses were performed using Sigmaplot v12.0 (Systat Software Inc., Chicago, IL, USA), and statistical significance was determined when the *p*-value was less than 0.05. Experimental results are expressed as mean ± standard deviation (SD). To test whether the variables were normally distributed, the Shapiro–Wilks test was used. If the variable distribution does not meet the normality, then the Kruskal–Wallis test is carried out. Otherwise, one-way ANOVA is used. Dunn test is used as the post hoc of the Kruskal–Wallis test, and Tukey’s method is used as the post test of one-way ANOVA.

## 3. Results

### 3.1. Network Pharmacology Analysis of GR on Compound-Disease Network

As a result of screening active compounds of GR using TCMSP, 69 compounds were identified (Appendix A). Based on the above analysis, a compound–disease network was constructed to visualize important compounds and diseases in the network (Figure 2). It was confirmed that among 69 compounds, MOL003896 (7-methoxy-2-methyl isoflavone), MOL000392 (formononetin), and MOL005003 (licoagrocarpin) interact the most with diseases (Figure 2A and Appendix A) that means these compounds can be used for treating various diseases.

Figure 2B shows notable interactions between diseases and compounds. Many compounds contained in GR, expected to have pharmacological activity, are highly likely to act on diseases, such as myocardial infarction, cardiovascular disease, and Alzheimer’s, and in addition, many compounds act on brain injury, including stroke, neurodegenerative diseases, and various inflammatory diseases. The frequency degree with which the compound is connected to disease in target–disease networks (Figure 2) is shown in Appendix A.

### 3.2. Effects on Damaged Brain Area and Neurobehavioral Changes

Ten brain tissue sections (1 mm) were obtained from the brain, excluding the olfactory bulb and cerebellum. TTC staining was used to acquire optical images of the damaged area and to identify the edema area (Figure 3A) and the infarct volume (Figure 3B). No edema or cerebral infarction was observed in the sham group. In contrast, a relatively extensive area of damage was observed in the model group. A significant reduction in brain edema was observed in mice administered 125 or 250 mg/kg body weight concentration of GRex (*p* = 0.031 and *p* = 0.046, respectively), and cerebral infarction was significantly reduced (*p* = 0.042) at the concentration of 125 mg/kg GRex. The cerebral infarction tended to increase when the dose concentration was increased to 500 mg/kg and 1000 mg/kg of GRex. In TTC-stained images, dot-like micro bleedings were observed (Appendix A), and it is presumed that these micro bleedings increased brain edema and infarction.

When glycyrrhizic acid, a major and standard compound of GR, was administered at a concentration of 3.75 mg/kg corresponding to the concentration of 125 mg/kg of GRex, there was no difference in the degree of cerebral infarction and edema compared to the model group (Figure 3A,B). This suggests that glycyrrhizic acid is not involved in the brain damage inhibitory effect of GRex.

### 3.3. Effects of GRex on Changes of the Forepaw Grip Strength and Behavior Tests

No statistical significance was observed in NDS among MCAO-induced groups (Figure 4A); however, since the NDS tended to decrease in the GRex groups with reduced cerebral infarct area, it is considered that a significant change may be seen if the long-term observation is performed after MCAO induction and adequate concentrations of GRex administration. No change in forepaw grip strength in the GRex 125 mg/kg group was observed, but a slight recovery trend was observed (Figure 4B), so long-term observation is required.

Figure 4C shows the routes taken by the experimental mice, and Figure 4D shows the total distance of spontaneous motor activities of the mice within the arena. The total distance within the arena was 1352.63 ± 196.88 cm in the sham group, 702.23 ± 193.89 cm in the MCAO model group, and 1137.33 ± 252.07 cm in the GRex 125 group. Total distance in the arena was significantly (*p* < 0.001) lower in the model group compared to the sham group, and the GRex 125 group showed a significant (*p* = 0.008) increase in total distance in the arena compared to the MCAO model group.

DR is one of the commonly used measurement methods for behavioral analysis in NORT [24,25]. The DR was 0.68 ± 0.10 in the sham group, 0.47 ± 0.08 in the MCAO model group, and 0.61 ± 0.06 in the GRex 125 group (Figure 4E); thus reducing the cognitive function of experimental mice due to MCAO-induced cerebral infarction was suppressed by administration of GRex.

### 3.4. Effects of GRex on Inflammatory Changes in Ipsilateral Cerebral Cortices

NeuN is a neuronal nuclear antigen commonly used as a biomarker for neurons. GFAP is one of the best markers for the activation of astrocytes following injury or stress in the central nervous system (CNS) [26,27,28]. Immunofluorescent staining of NeuN for neurons and GFAP for astrocytes was performed 24 h after MCAO. NeuN staining of the MCAO model group was weaker than that of the sham-operated group. In contrast, GFAP was stained more clearly (Appendix A), indicating that astrocytes were activated by MCAO-induced brain injury. This tendency was significantly reduced when GRex was administered (Figure 5A,B).

CD68 is a lysosomal protein expressed at high levels by macrophages and activated microglia at low levels by resting microglia. Microglial activation is observed in CNS diseases and is important for coordinating immune system resources during disease-associated neuroinflammation [29,30]. TNF-α is an inflammatory cytokine produced by macrophages or monocytes during acute inflammation and is involved in various signaling events within cells leading to necrosis or apoptosis [31,32]. As a result of immunofluorescent staining, the cells showing a CD68+ and TNF-α expression in the MCAO model group were significantly increased compared to the sham group. When the GRex extract was administered, the expression of TNF-α was not affected, but CD68+ cells appeared to be significantly reduced (Figure 5C–E and Appendix A).

The above results indicate that the inflammatory response was increased by the activation of cells involved in the inflammatory response in the brain, such as astrocytes or microglia, by MCAO-induced ischemia, and the administration of GRex significantly reduced this response.

### 3.5. Influence of GRex on NF-κB and TLR4 Proteins

NF-κB is considered a regulator of innate immunity, and acute cerebral ischemia induces an innate immune response, resulting in a cascade of events leading to necrotic death of neurons and damage to supporting structures of neurovascular units. NF-κB activation is initiated when molecules such as TNF-α bind to TNF receptors, and the induction of NF-κB target genes requires phosphorylation of NF-κB proteins, such as p65; therefore, factors that modulate the activity of NF-κB may modulate the inflammatory process in ischemic stroke [33,34,35]. TLR4 activation is associated with proinflammatory cytokine expression and activation of the NF-κB signaling pathway, and TLR4 expression is positively correlated with TNF-α levels [36,37]. In the current study, MCAO induction increased the expression of p-NF-κB and TLR4 proteins (*p* = 0.007 and *p* < 0.001, respectively) in the ipsilateral hemispheric brain. GRex administration with 125 mg/kg downregulated the expression of p-NF-κB and TLR4 proteins (*p* = 0.031 and *p* < 0.001, respectively) (Figure 6A,B).

## 4. Discussion

Ischemic stroke is a neurovascular disease caused by oxidative damage to the cells that make up the brain tissue and inflammation in the cranial nervous system, leading to motor and language impairment or death in severe cases. Ischemic stroke occurs due to blockage of blood vessels that supply the brain and accounts for 87% of all stroke cases [1,2,3,4]. With the increasing prevalence of metabolic diseases and significant risk factors for stroke, stroke prevention and treatment is an important area of medical research. Therefore, research into therapeutic agents that can prevent and treat stroke is essential [1,3,38].

The only FDA-approved treatment for ischemic stroke is tissue plasminogen activator, also known as tPA; tPA works by dissolving blood clots and improving blood flow to areas of the brain that lack blood flow. However, intravenous (IV) tPA administration is associated with increased intracranial hemorrhage (ICH) and hemorrhagic transformation. In particular, delayed tPA administration increases the risk of edema, hemorrhagic transformation, and ICH [39,40,41,42,43,44]. Therefore, there is still a need to develop drugs that can be safely used for ischemic stroke with few side effects other than tPA.

As described above, among the prescriptions used for ‘wind’-related diseases in Donguibogam, the frequency of use of GR was higher than that of other drugs. Network pharmacology analysis confirmed that GR could be effectively used for various brain diseases (Figure 2). The medicinal effects of GR first appeared in Shennongbencaojing (Shennong’s Classic of Materia Medica), were formed through herbal classics for a long time as following: governing ‘five viscera’ and ‘six vowels’, clears ‘heat’ and relieves ‘fire toxicity’, strengthens muscle and bones, nourishes flesh, tonifies powers, treats sore and swelling, and lightens one’s body and extends one’s life if it was taken for a long time [45]. Although little has been written about the side effects of GR in traditional Chinese or Korean medicine, side effects of GR have been reported through recent studies. Occasionally, GR ingestion causes pseudoaldosteronism as a side effect that causes edema, hypokalemia, and hypertension due to the hyperactivity of mineralocorticoid receptors [46,47]. Despite these side effects, GR is considered one of the most important herbal medicines that can reduce toxicity and increase the effectiveness of certain herbal medicines. Various pharmacological activities of GR have been reported through studies by many researchers that GR has a variety of beneficial effects, including treating throat infections, tuberculosis, respiratory and liver ailments, antibacterial, anti-inflammatory, and immunodeficiency [48,49]. However, research on whether GR can be effectively used for ischemic stroke is still insufficient. Although there has been a study that glycyrrhizic acid, the representative compound of GR, is effective in cerebral infarction in mice [17], new studies were required because of its low similarity to the current study using a different time window of GR administration. We have recently reported study results showing that pretreatment with GRex in experimental animals reduces cerebral infarction, but research on post-treatment is still lacking.

The recommended daily dose of GR was about 12 g/60 kg body weight in humans, and the corresponding dose of GRex for mice, considering the yield of GR to GRex and the metabolic rate of mice [50] compared to a human, was approximately 250 mg/kg. Therefore, the dosage showing pharmacological activity in the ischemic stroke mice model was searched for by increasing or decreasing the dosage based on 250 mg/kg GRex dosage. As a result, it showed the activity of significantly reducing cerebral edema and infarction at 125 mg/kg (Figure 3). However, when a 3.75 mg/kg dose of glycyrrhizic acid, the corresponding content at 125 mg/kg GRex, was administered, no pharmacological activity was observed, which indicates that glycyrrhizic acid is not involved in the action of GRex to reduce cerebral infarction.

When the dose of GRex was increased to 500 mg/kg and 1000 mg/kg, the area of cerebral infarction was significantly increased. When the tissues were observed by staining with TTC, dot-like micro bleedings were observed (Figure 3B and Appendix A). These results suggest that micro bleedings may be possible due to the administration of GRex. At least the ischemic stroke model used in this study may have been converted into hemorrhagic transformation. Shin et al. reported cerebral hemorrhage and micro bleeding as blood pressure increased in the elderly who received health supplements containing GR [51]. Although the evidence that intracranial micro bleeding occurs due to the administration of GR is still lacking, it is possible to speculate that GR may be related to intracranial micro bleeding based on the above report. Spontaneous motor activities and DR in the arena for NORT were reduced by MCAO-induced brain injury, and the administration of GRex ameliorated such changes (Figure 4). The results are presumed to be caused by the reduction of the cerebral infarct area. Microglia and astrocytes are key regulators of the inflammatory response in the CNS. Neuroinflammation is the activation of the immune response in the CNS by microglia and astrocytes, which indigenous immune cells, astrocytes, and microglia lead to the secretion of cytokines, including TNF-α, IL-1β, and IL-6 initiating inflammatory responses; these processes also occur in ischemic stroke [26,27,28,29,30,31,32]. This study activated astrocytes and microglia, and TNF-α was significantly overexpressed by MCAO induction. GRex administration downregulated astrocyte and microglia activation (Figure 5), indicating that the MCAO-induced inflammatory response was significantly reduced by GRex administration. The down-regulation was caused by inhibiting the activation of microglia and astrocyte.

Since cerebral ischemia induces innate immune responses and NF-κB is a regulator of innate immunity, agents that modulate the activity of NF-κB may modulate the inflammatory process in ischemic stroke, suggesting that TLR4 activation is associated with the expression and activation of proinflammatory cytokines of the NF-κB signaling pathway [33,34,35,36,37]. In the current study, GRex administration with 125 mg/kg downregulated the expression of p-NF-κB and TLR4 proteins (Figure 6), indicating that GRex administration could modulate the NF-κB signaling pathway.

In summary, GR, the most frequently used herbal medicine in prescriptions to treat stroke-related diseases in Korean medicine, could be used for brain diseases through network pharmacological analysis. GRex, the methanolic extract of GR, was post-treated with MCAO-induced ischemic stroke in mice to observe whether GRex post-administration is effective in cerebral infarction. As a result, we found that cerebral infarction was significantly reduced at 125 mg/kg dosage. The process of reducing inflammatory response by regulating the activation of astrocytes and microglia was involved as an underlying mechanism (Figure 7).

## 5. Conclusions

In order to investigate the neuroprotective effects of GR on stroke, GRex, a methanol extract of GR, was administered to mice with ischemic stroke. The MCA was occluded for 90 min to induce ischemic stroke, and several biological indicators of stroke were evaluated. GRex post-treatment significantly reduced edema area and infarct volume. GRex significantly inhibited the decrease of spontaneous motor activities and recognition function. The underlying action mechanism of GRex reducing inflammatory response was identified as regulating the activation of astrocytes and microglia.

## Figures and Tables

**Figure 1 pharmaceutics-14-02776-f001:**
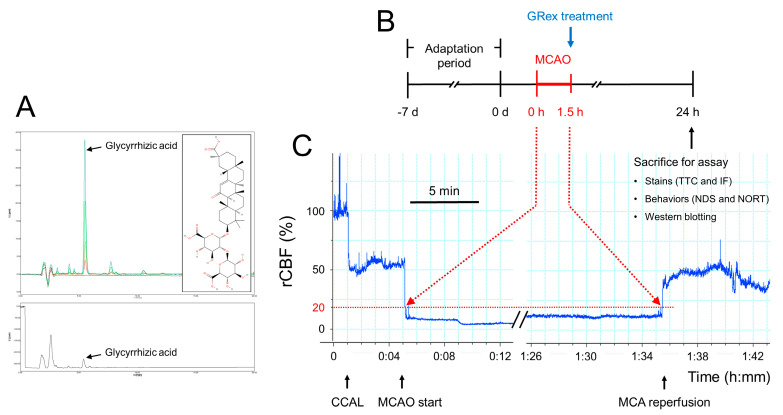
High-performance liquid chromatography (HPLC) analysis and experimental design for middle cerebral artery occlusion (MCAO) mice model: (**A**) HPLC chromatogram of the glycyrrhizic acid standards and its chemical structure (upper panel) and the methanolic extract of GR (GRex) (lower panel); conditions used: HPLC, YL-9100 system; column, ZORBAX Eclipse XBD-C18; mobile phase, a mixture of acetonitrile and 1% acetic acid in water (40:60, *v*/*v*); wavelength, 254 nm; column temperature, 25 °C; flow rate, 1.0 mL/min; injection volume, 10 μL; (**B**,**C**) ischemic stroke model in mice; mice were adapted for seven days at animal facilities. The MCAO was maintained for 90 min, and it was confirmed that relative cerebral blood flow (rCBF) was continuously maintained under 20%. After MCAO, GRex was administered, and biological indicators of experimental animals were measured at 22.5 h after reperfusion; CCAL, common carotid artery ligation.

**Figure 2 pharmaceutics-14-02776-f002:**
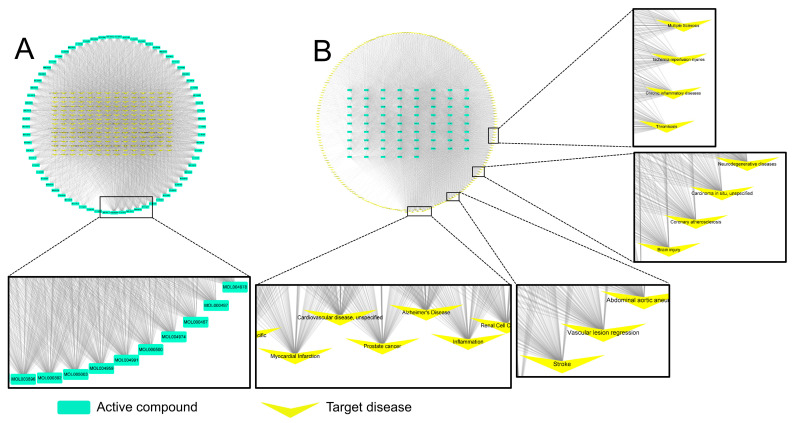
Visualization of compound–disease networks of *Glycyrrhizae Radix et Rhizoma* (GR). The light green nodes (rectangle-shape) represent the active compounds of GR, and the yellow nodes (V-shape) represent diseases. The gray lines represent the interaction of each compound and its target diseases: (**A**) the yellow inner nodes represent diseases, and the light green outer nodes represent active compounds of GR; (**B**) the inverted form of network A.

**Figure 3 pharmaceutics-14-02776-f003:**
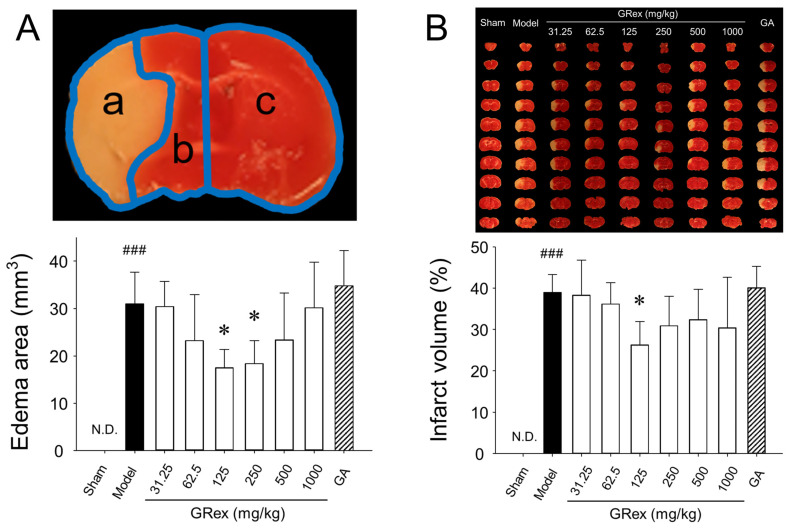
Measurements of edema area and infarct volume: (**A**) quantitative analysis of the edema area, determined as (a [ipsilateral infarct volume] + b [ipsilateral non-infarct volume]—c ([contralateral volume]) using 2,3,5-triphenyltetrazolium chloride (TTC)-stained brain slice; (**B**), representative photographs of TTC-stained brain slices (1 mm) showing the infarct area 24 h after middle cerebral artery occlusion (MCAO) (upper column) and quantitative analysis of the total infarct volume (lower column). GA, glycyrrhizic acid. All data are expressed as mean ± SD (*n* = 5). ### *p* < 0.001 compared to sham group and * *p* < 0.05 compared to MCAO model group.

**Figure 4 pharmaceutics-14-02776-f004:**
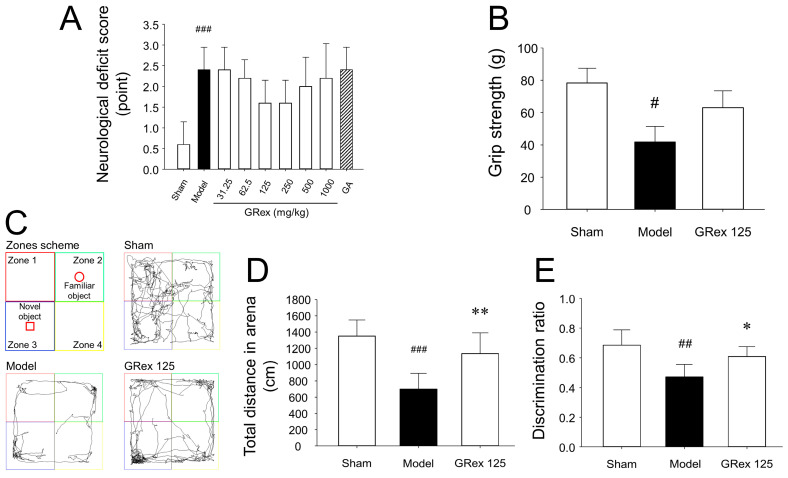
Measurements of neurological deficit scores (NDS), grip strength, and novel object recognition test (NORT): (**A**) NDS; (**B**) forepaw grip strength; (**C**) schematic view of NORT and the route taken by experimental mice; (**D**) spontaneous active movement during NORT; (**E**) discrimination ratio (DR). All data are expressed as mean ± SD (*n* = 5). # *p* < 0.05, ## *p* < 0.01, and ### *p* < 0.001 compared to sham group and * *p* < 0.05 and ** *p* < 0.01 compared to MCAO model group.

**Figure 5 pharmaceutics-14-02776-f005:**
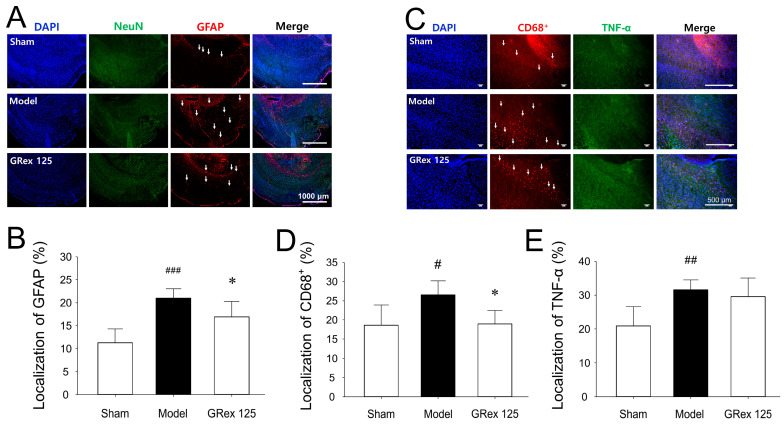
Attenuating neuronal inflammatory response in the cortical area of ischemic ipsilateral hemispheres. Each photomicrograph represents immunofluorescence (IF)-stained cortical regions: (**A**,**C**) represent IF-stained images of each group; (**B**,**D**,**E**) bar graphs represent mean ± SD (*n* = 5, per group) of IF color intensities of each antibody. White arrows indicate activated glial cells. All data are expressed as mean ± SD (*n* = 5). # *p* < 0.05, ## *p* < 0.01, and ### *p* < 0.001 compared to sham group and * *p* < 0.05 compared to middle cerebral artery occlusion (MCAO) model group.

**Figure 6 pharmaceutics-14-02776-f006:**
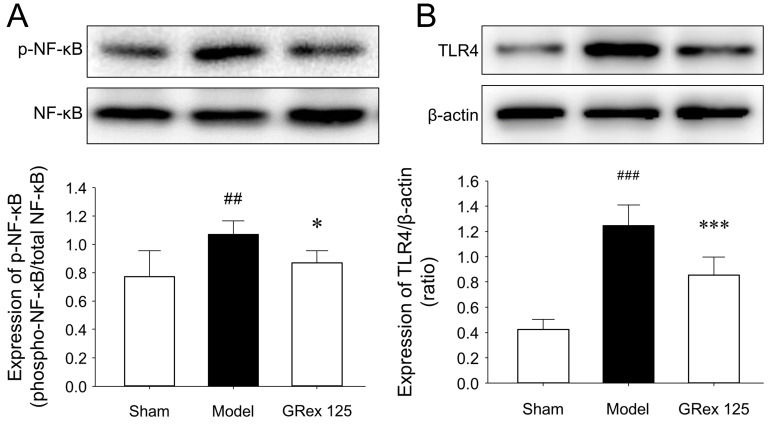
Effects of GRex post-treatment on the changes in NF-κB and TLR4 proteins in MCAO-induced mouse brain. Representative images (upper column of (**A**,**B**)) and relative densitometry (lower column of (**A**,**B**)) of Western blot analysis of expression levels in brain tissues (*n* = 5, per group). All data are expressed as mean ± SD (*n* = 5). ## *p* < 0.01 and ### *p* < 0.001 compared to sham group and * *p* < 0.05 and *** *p* < 0.001 compared to MCAO model group.

**Figure 7 pharmaceutics-14-02776-f007:**
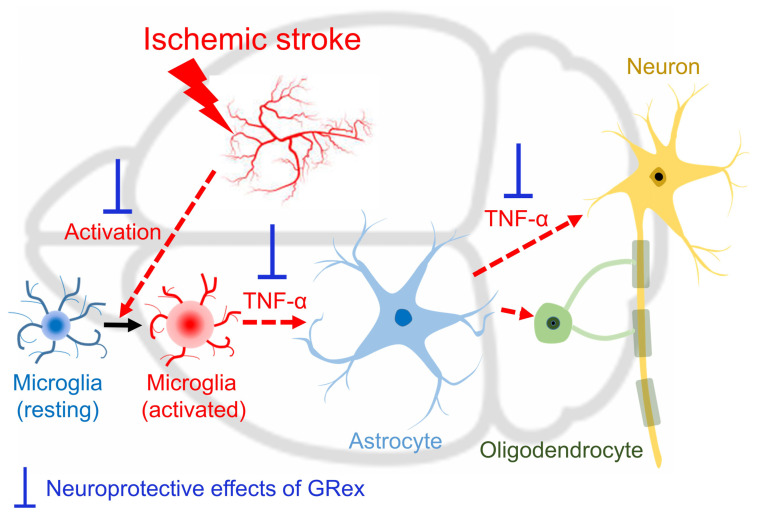
Schematic image of the anti-inflammatory action mechanism of GRex in MCAO mice model. Thickened brown bars indicate the predicted action site of GRex post-treatment on ischemia/reperfusion-induced brain damage.

## Data Availability

All data in this study are available from the corresponding author on reasonable request.

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
