# Peer review of "Amelioration of Brain Damage after Treatment with the Methanolic Extract of Glycyrrhizae Radix et Rhizoma in Mice"

_pharmaceutics, 2022, doi:10.3390/pharmaceutics14122776_

Round 1

Reviewer 1 Report

1.The samples of each group is too small, reducing the significance of the results.The conclusion drawn in this case is lack of reliability.

2. control group should be written as model group.

3.Figures are not clear, such as Figure 5.

4.The results said that "The neuroprotective effects of GRex on is- 25 chemic stroke were due to its regulation of inflammation-related neuronal cells, such as microglia 26 and astrocytes", but there is not enough data to prove it, there is no significant between control group and GRex 125 (TNF-a).

5.Figure 7 need to be improved.

Author Response

Comment 1. The samples of each group is too small, reducing the significance of the results.The conclusion drawn in this case is lack of reliability.

-> Thank you for your suggestion. Because a relatively large number of mice had to be used to test the activity of various concentrations of licorice extract, that’s why the number of mice in each group was not sufficient. Since the detailed mechanism of action at the bioactive concentration (125 mg/kg bw) will be studied in the follow-up study, we plan to use a sufficiently large number of animals in future studies.

Comment 2. control group should be written as model group.

-> Thank you for your suggestion. We have corrected it in the revised manuscript.

Comment 3. Figures are not clear, such as Figure 5.

-> Please refer to Supplementary Figures (Fig. S2 and S3).

Comment 4. The results said that "The neuroprotective effects of GRex on ischemic stroke were due to its regulation of inflammation-related neuronal cells, such as microglia and astrocytes", but there is not enough data to prove it, there is no significant between control group and GRex 125 (TNF-a).

-> Although licorice extract had no effect on TNF-a levels, it was described as such because it suppressed the expression of cells involved in inflammation and also reduced the area of cerebral infarction. As you mentioned above, we speculate that significant changes in TNF-a values may not have been observed because the number of mice used in our study was small. Therefore, in our follow-up study, which will identify various mechanisms of action related to inflammation, we would like to use a sufficient number of mice. Thank you very much for your kind and constructive comments.

Comment 5. Figure 7 need to be improved.

Thank you for your suggestion. We have modified the Figure 7 in the revised manuscript.

Reviewer 2 Report

In this manuscript Choi et al. describe the protective effect of methanolic extract of Glycyrrhizae Radix et Rhizoma in a mouse model of ischemic stroke. All results are coherent and very nice, and the experiments are carried out coherently.

Authors could add in discussion section about the potential active compound/s carrying the total effect of extract.

Author Response

Comment 1. Authors could add in discussion section about the potential active compound/s carrying the total effect of extract.

-> When glycyrrhizic acid, the most representative indicator of licorice, was pre-treated at a high concentration, it showed an activity to inhibit cerebral infarction. But licorice extract was post-treated in this study and it was confirmed that glycyrrhizic acid was not involved in the neuroprotection of licorice, although it is not known whether it was because of the low concentration.

For this reason, we concluded that there is still insufficient evidence to describe the speculation about which compound among the chemical ingredients of licorice have shown activity. However, in our follow-up study, we plan to use single ingredients as materials to determine whether they exhibit activity.

We would like to thank you for your important comments on setting the direction of our study.

Round 2

Reviewer 1 Report

It could be accepted after check it carefully.